# Using a network of temperature lidars to identify temperature biases in the upper stratosphere in ECMWF reanalyses

Graeme Marlton[1], Andrew Charlton-Perez[1], Giles Harrison[1], Inna Polichtchouk[2], Alain Hauchecorne[3], Philippe Keckhut[3], Robin Wing[3], Thierry Leblanc[4], and Wolfgang Steinbrecht[5]

[1]Department of Meteorology, University of Reading, Reading, RG6 6LA
[2]European Centre for Medium Range Weather Forecasts, Shinfield Road, Reading, United Kingdom
[3]LATMOS/IPSL, UVSQ Université Paris-Saclay, Sorbonne Univerités, CNRS, Guyancourt, France
[4]JPL-Table Mountain Facility, 24490 Table Mountain Road, Wrightwood, CA., USA
[5]Deutscher Wetterdienst, Albin-Schwaiger-Weg 10, 82383, Hohenpeissenberg, Germany

**Correspondence:** Graeme Marlton (graeme.marlton@reading.ac.uk)

**Abstract.** To advance our understanding of the stratosphere, high quality observational datasets of the stratosphere are needed. It is commonplace that reanalysis datasets are used to conduct stratospheric studies. However the accuracy of these reanalyses at these heights is hard to infer due to a lack of in-situ measurements. Satellite measurements provide one source of temperature information. As some satellite information is already assimilated into reanalyses, the direct comparison of satellite temperatures to the reanalysis is not truly independent. Stratospheric lidars use Rayleigh scattering to measure density in the middle and upper atmosphere, allowing temperature profiles to be derived for altitudes from 30km (where Mie scattering due to stratospheric aerosols becomes negligible) to 80-90km (where the signal-to-noise begins to drop rapidly). The Network for the Detection of Atmospheric Composition Change (NDACC) contains several lidars at different latitudes that have measured atmospheric temperatures since the 1970s, resulting in a long running upper-stratospheric temperature dataset. These temperature datasets are useful for validating reanalysis datasets in the stratosphere, as they are not assimilated into reanalyses. Here we take stratospheric temperature data from lidars in the northern hemisphere between 1990-2017 and compare them with the European Centre for Medium range Weather Forecasting ERA-interim and ERA5 reanalyses. To give confidence in any bias found, temperature data from NASA's EOS Microwave Limb Sounder is also compared to ERA-interim and ERA5 at points over the lidar sites. In ERA-interim a cold bias of -3 to -4 K between 10 hPa and 1 hPa is found when compared to both measurement systems. Comparisons with ERA5 found a small bias of magnitude 1 K which varies between cold and warm bias with height between 10 hPa and 1 hPa, indicating a good thermal representation of the middle atmosphere up to 1 hPa. A further comparison is undertaken looking at the temperature bias by year to see the effects of the assimilation of the Advanced Microwave Sounding Unit-A satellite data and the Constellation Observing System for Meteorology, Ionosphere, and Climate GPS Radio Occulation (COSMIC GPSRO) data on stratospheric temperatures. It if found that ERA5 is sensitive to the introduction of COSMIC GPRSO in 2007 with the reduction of the cold bias above 1 hPa. In addition to this, the introduction of AMSU-A data caused variations in the temperature bias between 1-10 hPa between 1997-2008.

## 1 Introduction

The stratosphere influences the weather and climate in the troposphere (Domeisen, 2019; Domeisen et al., 2019a). Sudden Stratospheric Warmings (SSWs) can cause changes in the tropospheric flow for many weeks (Charlton and Polvani, 2007) and the Quasi Biennial Oscillation (QBO), a 28 month switching in equatorial stratospheric winds, also affects the large scale processes in the troposphere (Baldwin et al., 2001). Critical to our understanding of how these processes work are good observational datasets of the middle atmosphere. Often reanalysis datasets such as the European Centre for Medium range Weather Forecasts' (ECMWF) ReAnalysis (ERA) (Dee et al., 2011) or the US National Center for Atmospheric Research (NCAR) ReAnalysis (Kalnay et al., 1996) amongst many others are used for stratospheric studies on a global scale as shown for example in Fujiwara et al. (2017); Seviour et al. (2012); Lee et al. (2019); Skerlak et al. (2014); Butler et al. (2015).

To create reanalyses datasets, a large amount of temperature, ozone and wind observations are assimilated from satellite and in-situ measurements. In the middle and upper stratosphere the number of temperature observations is somewhat limited. This makes diagnosing bias in a reanalysis dataset more difficult. Radiosondes, small balloon borne instrument packages, which provide in-situ temperature profiles up to heights of about 30 km are launched from thousands of locations daily, giving a wealth of information that is assimilated in the lower and middle stratosphere. Simmons et al. (2020) undertook a study examining the performance of ERA5, the ECMWF's most recent reanalysis dataset using radiosonde and satellite observations. However, there were height limitations due to the maximum height of available radiosonde data. The technique also had the potential to be biased due to the assimilation of radiosonde observations into the reanalysis. Rocketsondes (Schmidlin, 1981) can reach heights of 100 km providing high resolution temperature information in the upper stratosphere, mesosphere and thermosphere. Rocketsondes are not operationally assimilated due to the significant installations and costs to launch, which has led to sparse temporal and spatial sampling, with the last known campaign occurring in 2004 (Sheng et al., 2015).

There are numerous satellite techniques to retrieve temperature profiles of the stratosphere. Stratospheric sounding units (SSU) (Miller et al., 1980) derive temperature from radiances in $CO_2$ emissions. Similarly the Sounding of the Atmosphere using Broadband Emission Radiometry (SABER) instrument uses limb emissions from $CO_2$ to provide temperature observations in the mesosphere and thermosphere (Russell et al., 1999). The Aqua satellite combines data from Atmospheric Infra-red Sounders (AIRS) with data from the Advanced Microwave Sounding Units (AMSU) to provide temperature profiles in the troposphere and stratosphere (Susskind et al., 2006). The Microwave Limb Sounder (MLS) provides temperature data by observing the limb emission of several atmospheric gases and aerosols (Waters et al., 2006). Low Earth orbiters such as those in the Constellation Observing System for Meteorology, Ionosphere and Climate (COSMIC) can derive atmospheric properties such as temperature, pressure and water vapour using GPS Radio Occulation (GPSRO) up to 40-50 km (Kuo et al., 2000). As COSMIC is a constellation of satellites it retrieves thousands of randomly sampled temperature profiles daily across the globe. Many of the above observations are assimilated into reanalyses making it hard to make an unbiased comparison.

Another source of stratospheric temperature measurements is from Rayleigh temperature lidars. These temperature lidars use Rayleigh scattering properties of the atmosphere above the stratospheric aerosol layer (>30 km) to infer density and, by assuming hydrostatic equilibrium, temperature (Hauchecorne and Chanin, 1980). There are several Rayleigh lidars across the globe based at participating sites in the Network for the Detection of Atmospheric Composition Change (NDACC). A small handful have been making temperature profile measurements between 30 km and 90 km for at least three decades. Furthermore the lidar temperature profiles are not assimilated into reanalyses making them independent for numerical dataset comparisons. Le Pichon et al. (2015) compared 6 months of Rayleigh temperature lidar data with ECMWF reanalysis data and found good agreement.

In this paper, we compare temperature data from four ground based Rayleigh lidars, an independent measurement technology, and use them to infer the stratospheric temperature bias in the ECMWF's ERA-interim and ERA5 reanalyses. To add confidence to the identified bias the same comparison will also be undertaken with temperature data from the National Aeronautics and Space Administrations (NASA) Earth Observing System Microwave Limb Sounder (EOS MLS). EOS MLS is one of the few satellite temperature datasets which is not assimilated into ECMWF reanalysis.

The reanalysis packages cover long time periods over which the quantity and types of data assimilated has increased. Poli et al. (2010) showed that inclusion of GPSRO data improved temperature bias in the lower stratosphere and upper troposphere. Simmons et al. (2020) also found that the inclusion of AMSU-A data in 2000 caused an increase in the warm bias in ERA5 at heights above 7 hPa. The four temperature lidars used here span at least 25 years. Hence, further analysis will be undertaken to ascertain how ERA-interim and ERA5's stratospheric temperature bias evolve over the period 1990-2017 with the introduction of both COSMIC GPSRO and AMSU-A data.

## 2 Dataset description

### 2.1 Stratospheric Temperature Lidar

Atmospheric lidar remote sensing uses light scattering from molecules and particles. A laser pulse is emitted into the atmosphere where it is scattered and absorbed by the molecules and particles. The fraction of light backscattered towards the instrument on the ground is collected by a telescope and sampled as a function of time, which, knowing the speed of light, translates into altitude. In the absence of particulate matter (typically above 30-35 km), and after several corrections (e.g., non-linearity and range corrections, background noise extraction, molecular extinction and absorption corrections), the lidar signal is proportional to the air density. With the assumption that the atmosphere is an ideal gas in hydrostatic balance, the atmospheric temperature profile is then retrieved by integrating the measured relative density downward from the highest usable data point (Hauchecorne and Chanin, 1980). At the top of the profile (typically, in the mesosphere), a priori temperature, density or pressure information is needed to initialize the downward integration.

At mesospheric altitudes, empirical models such as the Committee on Space Research's International Reference atmosphere (CIRA-86) (Chandra et al., 1990) or the Naval Research Laboratory's Mass Spectrometer and Incoherent Scattering Radar model (NRL MSISE-00) (Picone et al., 2002) are typically used for the a priori information. Because of tides and gravity waves,

mesospheric temperatures can be highly-variable at small spatio-temporal scales (Jenkins et al., 1987), and these models often do not represent well the state of the atmosphere measured by lidar at a given time and location, resulting in a temperature uncertainty of up to 10-20 K at the altitude of initialization. This uncertainty decreases exponentially as the profile is integrated downward, resulting in a typical temperature uncertainty of 1-2 K 15 km below the top of the profile due to the a priori information (Leblanc et al., 2016). In order to avoid misinterpretation of the lidar profile and its influence by the a priori information, the top 10 km of the profiles are often excluded from published datasets (Wing et al., 2018).

Another important source of temperature uncertainty is signal detection noise which has two components. The first is a random component in the form of photon detection, which becomes negligible when averaging, and a systematic component in the form of background noise such as level of sky light levels etc., which can be budgeted and corrections applied for (e.g. Leblanc et al. (2016)). At the bottom of the profile the random temperature uncertainty is negligible as the signal-to-noise ratio is high, but can increase to 10 K at the top of the profile where signal-to-noise ratio decreases. Because of its random nature, vertical and temporal averaging can reduce detection noise significantly. Background noise correction and signal non-linearity correction are two other sources of uncertainty. Just like a priori and detection noise uncertainty, the corrections for background noise uncertainty is negligible at the bottom of the profile and increases as we approach the top of the profile. On the other hand, uncertainty owed to non-linearity correction maximizes at the bottom of the profile (typically less than 2 K), and becomes negligible a few km above.

It should be stressed that each lidar instrument is different and the various contributions to total uncertainty can vary widely depending on the instrument considered (Leblanc et al., 2016). For the comparisons undertaken here, we selected the four longest datasets of the dozen backscatter temperature lidar datasets available at NDACC. These datasets span at least 25 years and have frequent temperature profiles during that period (see table 1). The instruments are located at the German Weather Service Observatory of Hohenpeissenberg (HOH) (Steinbrecht et al., 2009; Wing et al., 2020a), Germany, the Observatoire de Haute Provence (OHP) (Hauchecorne, 1995; Wing et al., 2020b), France, the JPL-Table Mountain Observatory Facility (TMO) (Ferrare et al., 1995) and Mauna Loa Observatory (MLO) (Li et al., 2008), US. All four instruments have very similar power and performance specifications, and follow a similar mode of operation (a few hours per night, 1 to 4 times per week, weather permitting), making it easier to include into a consistent ground-based reference combined dataset. For these instruments, the temperature total uncertainty ranges from 2-3 K at 30 km to less than 1-2 K between 35 and 55 km, and then back up to 2-5 K in the mid-mesosphere, and up to 20 K near the initialization altitude (80-95 km).

Validating Rayleigh temperature lidar measurements in the upper stratosphere and mesosphere can be difficult due to lack of reference temperature observations at these altitudes. Occasional comparisons with rocketsonde measurements showed temperature differences of 2-5 K in the lower mesosphere (Schmidlin, 1981; Ferrare et al., 1995). Over the past two decades, the performance of Rayleigh temperature lidars has been evaluated mainly by comparison with satellite measurements during which they served as the ground-based reference. These inter-comparisons typically yield lidar-satellite differences not exceeding 2-4 K between 30 and 60 km (Wang et al., 1992; Ferrare et al., 1995; Wu et al., 2003; Sica et al., 2008; García-Comas et al., 2014; Stiller et al., 2012). At the bottom end of the profiles (below 30-35 km), the lidars have been compared to radiosonde measurements (Ferrare et al., 1995). In the presence of aerosols, Rayleigh backscatter lidars yield a significant cold bias (the

thicker the aerosols, the colder the bias). The TMF and MLO instruments include an additional vibrational Raman backscatter channel, i.e., unimpacted by aerosol backscatter, which allows temperature retrieval down to 10 km (Gross et al., 1997; Leblanc et al., 2016), but with a remaining slight cold bias (1-4 K) due to aerosol extinction. For these two lidars the reanalysis packages described in section 2.3 will be compared for greater altitude ranges than the HOH and OHP lidars.

To fill the need for additional validation, one method frequently used within NDACC is to deploy a mobile lidar from site to site to blind-test the lidar instruments permanently deployed at these locations. One such "travelling standard" is operated by the NASA Goddard Space Flight Centre (GSFC) (McGee et al., 1995) and has been used to validate the lidar instruments at HOH, OHP, TMF, and MLO (Ferrare et al., 1995; Leblanc et al., 1998; Keckhut et al., 2004; Wing et al., 2020a, b). During these campaigns, temperature differences between the lidars did not exceed 4 K in the 25-60 km altitude range, with typical
differences of +/-2 K.

## 2.2   Microwave Limb Sounder

The NASA EOS MLS was launched on the 14th July 2004 and became operational on the 14th August 2004. It works by observing millimetre and sub-millimetre thermal emission along the limb of the atmosphere. It is in a low polar orbit allowing it to orbit the Earth 15 times a day, crossing the equator near to local noon and midnight. It is designed to measure several
atmospheric gases and aerosols in the upper troposphere, stratosphere and mesosphere (Waters et al., 2006). It uses the emission from oxygen to provide temperature and pressure measurements; the precision of the measurement is given in Waters et al. (2006) to be 0.5-1 K between 300 and 0.001 hPa, where it has a vertical resolution between 4-8 km. Initial comparisons by Froidevaux et al. (2006) with other satellite retrievals of temperature found that EOS MLS had a 1-2 K warm bias. A more thorough comparison made by Schwartz et al. (2008) compared EOS MLS temperature retrievals to those from radiosondes
and several satellites, including GPSRO. It was found that from 0.0001 hPa to 0.3  hPa the temperature bias could range from -9 to 0K with temperature precision ranging from $\pm 1$ to $\pm 2.5$ K. A further study by Wing et al. (2018) found that the bias in wintertime MLS was -10 K and $\pm 4$ K in the summertime. At 1 hPa warm biases from 0 to 5K were found. From 3.16 hPa down to 316 hPa precision was found to be less than $\pm 1$ K and biases were between -2 and 1.5 K. Thus, at pressure heights of 3 hPa and below the EOS MLS satellite has a similar bias to that of the temperature lidar at the same observing height.
Wing et al. (2020b) compared EOS MLS at the OHP lidar against NASA's reference lidar and it was found to have a large cold bias above 3 hPa of -10 K. Figure 1 panels a-d shows the mean MLS and Lidar temperature profiles between 2004 and 2017 for HOH, MLO, OHP and TMO respectively. Panels e-h shows the average difference between the matched profiles at HOH, MLO, OHP and TMO respectively. This shows a cold bias which increases in magnitude with height and agrees well with the findings of Schwartz et al. (2008),Wing et al. (2018) and Wing et al. (2020b).

## 30   2.3   European Centre for Medium Range Forecasts data

A reanalysis dataset is generated by combining many different historical measurements using data assimilation to create an accurate numerical representation of the Earth's atmosphere at a given time. ERA interim spans from 1979 to 2019. ERA-interim has 60 model levels spanning the surface to 0.3 hPa (57 km altitude) with an approximate 79 km horizontal grid

resolution and 6 hour analysis windows (Dee et al., 2011) . It is based on the Integrated Forecasting System (IFS) cycle 31R2 and utilises a 4Dvar data assimilation system. Dee et al. (2011) state that during December 2006 GPSRO data from the COSMIC constellation was included in reanalysis datasets. (Poli et al., 2010) showed that the variability in ERA-interim temperature was much improved after this time.

ERA5 is the 5th generation reanalysis created by the ECMWF and replaces ERA-interim. The new ERA5 reanalysis, based on IFS cycle 41R2 (Hersbach et al., 2020), has 137 model levels from surface to 0.01 hPa (approximately 80 km) and a global horizontal resolution of 31 km, compared to ERA Interim's 60 model levels and 79 km horizontal resolution. Amongst 10 years of research and development due to the constant evolution of the ECMWF IFS are the inclusion of more measurement systems, improved bias correction techniques and model physics, CMIP5 radiative forcings, and data assimilation using a hybrid 4Dvar
system (Hersbach et al., 2020). Simmons et al. (2020) showed that temperature bias in the upper stratosphere of ERA5 was significantly effected with the addition of AMSU-A data between 2000 and 2007 at heights above 15 hPa.

## 3    ERA-interim comparisons

In this section MLS and lidar profiles will be compared with temperature profiles from ERA-interim. The lidar temperature profiles were interpolated onto ERA-interim's model levels using geopotential height $Z$, for time steps that were closest to the
mid-point of the lidar's profile acquisition period. To ensure the comparison is accurate the lidar's height coordinate which is in geometric height was first converted to geopotential height. Despite being similar near ground level, the differences are between 0.4 km and 1 km for the altitude ranges used in this study. The vertical resolution for ERA-interim at these heights is approximately 1.5 km at 10 hPa and 3 km at 1 hPa. For this comparison we use lidar and reanalysis profiles for the full temporal lidar temperature profile range shown in table 1. Figure 2 panels (a-d) shows matched mean temperature profiles from both lidar
(red) and ERA-interim (blue) for the lidar sites at HOH, MLO, OHP and TMO respectively. Panels e-h show the mean of the matched differences for the corresponding profiles (a-d); grey shading shows 1 standard deviation in the matched differences. ERA-interim at the points studied here has a cold bias in the region of -3 to -4 K. Crosses are red where the mean difference is different from zero at the 95% significance level. For HOH, OHP and TMO, the peak cold bias is centred between 10hPa and 1hPa. For MLO the cold bias extends down to the 100 hPa pressure surface, whereas at TMO the cold bias is much closer to
zero between 10 hPa and 100 hPa. A possible hypothesis is that TMO is at a higher latitude than the tropically positioned MLO where the representation of the middle atmosphere within ERA-interim differs slightly. For all sites ERA-interim exhibits a warm bias between 1 hPa and 0.1 hPa. This contrastingly warm bias is due to the model top being reached and the stratopause not being represented. Panels i-l show the seasonal variation of the temperature biases with height. The warm bias near the model top is present through out the year, the cold bias between 1 hPa and 10 hPa is present throughout the year, which
intensifies at all sites during the November to February period.

     A similar analysis was performed using the EOS MLS data. The EOS MLS temperature data was first sorted so that only night time passes within $2.5^o$ of each temperature lidar site were retained. The remaining EOS MLS temperature profiles were interpolated onto ERA-interim's model levels using $Z$ for time steps that were closest and less than 3 hours apart. Due to EOS

MLS only being launched in 2004 the results of the ERA-interim and MLS comparison shown in figure 3 are for the years 2004 to 2017. Figure 3 shows matched mean temperature profiles from both MLS (red) and ERA-interim (blue) for lidar sites at HOH, MLO, OHP and TMO respectively. Panels (e-h) show the mean of the matched differences for MLS over the 4 lidar sites. The cold bias seen in figure 2 is not present when comparing with EOS MLS, instead a warm bias is present of approximately

1 K from 100 hPa to 1 hPa at HOH, MLO and OHP. At TMO a cold bias of -1 to -2 K is observed at 1 hPa, the large warm bias above 1 hPa near the ERA-interim model top shown in figure 2 is also observed. MLS shows a different temperature bias to that of the temperature lidar. In figure 1 we see that MLS has a cold bias when compared to the lidar at HOH and OHP and also in Schwartz et al. (2008), Wing et al. (2018) and Wing et al. (2020b). As the MLS records a negative bias when compared with the Lidar, and ERA-interim also exhibits a cold bias compared to the lidar, we see a resulting warm bias in the ERA-interim

comparison with MLS. The cold bias seen at 1 hPa at TMO is due to the warm bias in MLS. The results seen here further demonstrates evidence of a systematic bias between the lidar and MLS measurement technologies at these heights. There is some oscillatory structure in the temperature bias that has been seen in Wing et al. (2018) who compared EOS MLS to lidar, and found that the oscillations are not retrieval level dependent. Furthermore the amplitude of the oscillations falls within the precision of EOS MLS given in section 2.2

Simmons et al. (2020) compared the global mean temperature from ERA-interim with global mean radiosonde data between 15 hPa and 7 hPa, and 7 hPa and above. Here our results in figure 2 agree with a cold bias seen in Simmons et al. (2020) for the height range 15-7 hPa, although the magnitude is much smaller. For the 7 hPa range and above our results disagree with Simmons et al. (2020) who found that there was a warm bias of 1-2 K whereas our results using the lidar a show a cold bias at these heights at HOH , MLO and TMO. The temperature bias shown in Simmons et al. (2020) are a global mean, whereas our

measurements are made at fixed locations, Moreover, the bias is calculated using radiosonde data which are not independent as they are assimilated into ERA-interim. This explains the polarity difference in the results shown in this study.

In brief conclusion, the temperature lidars have shown that a cold bias of -3 to -4 K exists between 1 and 10 hPa in ERA-interim for the HOH and OHP lidar sites and between 1 and 100 hPa for the TMO and MLO sites.

## 4   ERA5 comparisons

To compare the temperature biases in ERA5 with those in ERA-Interim we repeat the comparison for ERA5 in a similar manner. The temperature lidar data was interpolated, accounting for geometric height, onto ERA5 model levels using $Z$ for the nearest analysis window to the mid point of the lidar acquisition window. The vertical resolution at 10 hPa is 750 m and at 1 hPa is 1.6 km. Figure 4 panels (a-d) show the mean temperature profiles for the lidar in red and ERA5 in blue up to a height of 0.5 hPa for the period 1990-2017. At first inspection ERA5 profiles track the lidar profiles more closely than those

of ERA-interim, including a more accurate representation of the stratopause than that of ERA-interim (see figure 2). Figure 4 panels (e-h) show the mean of matched differences with height; grey shading shows 1 standard deviation in the matched differences and crosses are red where the mean difference is different from zero at the 95% significance level inferred by a single sample t-test. The temperature biases are significantly smaller than in ERA-interim. For the MLO and TMO sites the

temperature bias is very close to zero up to the 10 hPa pressure surface. At 10 hPa to 5 hPa a cold bias of -1 to -2 K is observed at all sites. From 5 hPa to 1 hPa the bias drops to near zero again and above 1 hPa a -3 K cold bias is observed at all four sites. When we consider that the measurement accuracy of the lidar is $\pm$ 2 K, ERA5 gives a good thermal representation of the atmosphere up to 1 hPa. Figure 4 panels (i-l) shows the seasonal variation of temperature bias with height. For all four sites there is a slight warm bias of approximately 1 k at 1-5 hPa during the summer months (May-August) with the exception of MLO where the warm bias at this height spans January to July. The cold bias above 1 hPa intensifies in the winter months.

Figure 5 shows the temperature comparisons against MLS for the period 2004 to 2017; the data was interpolated onto the ERA5 model levels using the same method as discussed in section 3. MLS and ERA5 show a fair agreement at all sites between 100 hPa and 10 hPa. But not as good as when compared to that of the lidar at TMO and MLO. From 100 hPa to 5 hPa there is a warm bias which varies with amplitude between 1-2 K. The warm bias peaks at 3 hPa with an amplitude of 3 K, at 1 hPa and above a large cold bias of -5 K is found. Panels (i-l) show that there is little seasonal variation in the temperature bias between ERA5 and MLS. Given the findings of Schwartz et al. (2008), Wing et al. (2018) and figure 1 it is clear that these bias are largely due to the MLS temperature bias previously discussed in section 2.2.

Simmons et al. (2020) compared the global mean temperature from ERA5 with global mean radiosonde data from 15 hPa upwards and found that performance was similar to ERA-interim pre 2000. A warm bias of 2 K was found above 7 hPa and a slight cold bias of less than -0.5 K between 7 hPa and 15 hPa for ERA5. Between 2000 and 2007 there was an increase in the warm bias making the bias 3 K and 0.5 K for the layers 7 hPa and above and 7-15 hPa respectively. Simmons et al. (2020) believed that this may be due to the introduction of observations from the AMSU-A instrument aboard NOAA 15 and NOAA 16 (Aumann et al., 2003) from 2000. Post 2007, after the introduction of GPSRO data, Simmons et al. (2020) showed the temperature bias above 7 hPa to be 1.5 K and less than 0.5 K between 7-15 hPa. As our comparisons using the lidar span 1990 to 2017 we do see some aspects of findings in Simmons et al. (2020) in that a warm bias is observed above 7 hPa during the summer months. In this section we have shown that ERA5 gives a much improved representation of the upper stratosphere when compared to ERA-interim. The focus has been on calculating temperature bias over a 1990-2017 and 2004-2017 study period for the lidar and MLS respectively. This means it is difficult to see if a particular data stream improved ERA-interim or ERA5's representation of the upper stratosphere. Given that the lidars have made measurements for over 25 years we can examine the data further by year to see how the introduction of new observation streams such as GPSRO and AMSU-A affected the ERA datasets.

## 5 ERA performance due to assimilation of COSMIC GPRSO and AMSU-A

The mean of differences between lidar and both ERA-interim and ERA5 were decomposed by year to examine if the introduction of new satellite data streams such as COSMIC GPSRO and AMSU-A changed the stratospheric temperature bias. Poli et al. (2010) showed that the inclusion of COSMIC GPSRO data improved reanalysis bias in the upper troposphere and lower stratosphere. However, their comparison was only undertaken between 200 hPa and 100 hPa. The effect of the inclusion of COSMIC GPSRO at 100 hPa was described in Cardinali and Healy (2014) who found that the introduction of COSMIC

GPSRO showed a decrease in forecast error between 250 hPa and 50 hPa. Simmons et al. (2020) showed a change in ERA5 temperature bias after the inclusion of the AMSU-A satellite data in 1999/2000. Data from MLS is only available post 2004 and given AMSU-A data became available 19998-2000 we restrict our time series analysis to the lidar data only.

Figure 6 shows the average annual temperature difference as a function of year and height between ERA-interim and the temperature lidar. HOH, OHP and TMO show an increase of the cold bias between 1 and 10 hPa until 1995-1996. It is not known if this occurred at MLO, due to lack of data over this period. Figure 14 in Dee et al. (2011) shows that during this 1995/1996 period the NOAA 14 SSU unit was launched and added to the data assimilation streams. This could explain the decrease in the cold bias at the 1 hPa to 10 hPa pressure range. During the 1998/1999 period a warm bias, similar to that experienced at the model top formed around the 1 hPa pressure level, which coincided with the inclusion of AMSU-A satellite data in 1998. Simmons et al. (2020) discussed how the addition of AMSU-A may have an affect on ERA-interim. It is apparent here it may also increase the warm-bias of ERA-interim at 1 hPa. HOH and TMO show subtle reductions in the cold bias from -5 to -4 in the 10-1 hPa range post 2007. With the exception of OHP which has an intensifying cold bias between 2014 and 2017 there are no further significant changes in temperature bias over the studied period.

Figure 7 shows the average annual temperature difference as a function of year and height between ERA5 and the temperature lidar. In the 3-5 hPa range at all sites there is a warm bias of 2-3 K between 1994 and 1997/1998. Given the abrupt and consistent reduction of the warm bias at all sites during 1998, the year AMSUA aboard NOAA-15 data began being assimilated, it is clear it reduced the warm bias at this height. However by 2000 the warm bias returned at this height range and is most dominant at OHP and TMO. One explanation could be that the addition of AMSUA aboard NOAA-16 which was ingested between the years of 2001-2009 (Hersbach et al., 2020) could be the reason for this warm bias, agreeing with the conclusions in Simmons et al. (2020). The cold bias at 1 hPa at HOH, MLO and TMO reduced from -2K to near 0 K at the end of 2006 which coincided with COSMIC GPSRO being made available for assimilation from December 2006. Although GPSRO has a tangent height of 50 km the assimilation of the bending angle means it can contribute observations at higher altitudes than 50 km (Healy, 2008). Additionally the hydrostatic nature of the model means that observations assimilated at a given level affect those above and below.

In summary, the cold temperature bias in ERA5 above 1 hPa is reduced from -2 K to near 0 K at 3 out of 4 of the sites post 2007 due to the inclusion of GPSRO data. Inclusion of the AMSU-A data on NOAA-15 from 1998 appears to reduce a warm bias at 3 hPa. This warm bias reappears at some sites with the introduction of AMSU-A data from NOAA 16. Although the instruments on both satellites are similar, inter satellite brightness temperature bias has been shown before by Mo (2011) between the AMSU-As on both NOAA-18 and NOAA-19 satellites. This could explain the opposing bias seen here. When comparing these findings to that of Simmons et al. (2020) who showed an intensifying of a warm bias between 2000 and 2007 globally we see agreement at OHP and TMO.

## 6 Conclusions

In this work we have utilised temperature lidar from the NDACC network, that are not assimilated into reanalyses, to identify temperature bias in the upper stratosphere in ERA-interim and ERA5 reanalyses. For comparisons with ERA-interim and the lidar, a cold bias of -3 to -4 K between 10 hPa and 1 hPa and a large warm bias above 1 hPa was found. The cold bias intensified in Northern hemisphere winter to -5 to -6 K. For ERA5 the temperature bias between the lidar and ERA5 is within $\pm 1$ K to a height of 1 hPa, which given the measurement accuracies of the lidar $\pm$ 2 K gives a good thermal representation of the stratosphere. Above 1 hPa a cold bias of -2 to -3 K is found. Similar to ERA-interim, ERA5 cold bias above 1 hPa intensifies to -4 K in the Northern hemisphere winter months and becomes a warm bias in the summer months. When comparing to MLS, ERA-interim exhibited a warm bias of 1 K. ERA5 had a warm bias of 1-2 K up to 5 hPa. Above this height MLS's warm bias at 3 hPa and the large cold bias at 1 hPa, both shown here in figure 1, Schwartz et al. (2008) and Wing et al. (2018) saturate the temperature bias results.

When examining the ERA-interim lidar comparison over 1990-2017 we see a warm bias increase at 1 hPa around 1997/1998, which could be due to the introduction of AMSU-A. There is also a small reduction from -5 K to -4 K post 2006 which is most noticeable at 1-2 hPa at HOH and TMO. This could be an indication that the inclusion of COSMIC GPRSO has an affect on the upper stratosphere within ERA-interim. For ERA5 the effects of new satellites being assimilated is clearer. We see that the inclusion of the two AMSU-A data streams had an effect on temperature bias between 1-10 hPa. It appears that in 1998 during the assimilation of AMSU-A from NOAA-15 the warm bias improved. Yet when AMSU-A from NOAA-16 was assimilated in 2000-2009 the warm bias returned. However it was not as intense. Post 2007 a reduction in the cold bias above 1 hPa was observed at HOH, MLO and TMO due to the assimilation of COSMIC GPSRO. Other small changes in the temperature bias seen in figures 6 and 7 could be attributed to other observations being assimilated. Both Hersbach et al. (2020) and Dee et al. (2011) show that in both ERA5 and ERA-interim the amounts and type of observations increase with time making it harder to characterise behaviours in the upper stratospheric temperature bias to a particular observation dataset.

From the comparisons here it can be stated that ERA5 is much improved compared to ERA-interim and has a good thermo-dynamic representation of the upper stratosphere. When we consider the uncertainties in the lidar ERA5 is an excellent choice for further stratospheric studies or for the use as reference to compare other reanalyses to. However, a cold bias detected in ERA5 by the lidar before the inclusion of GPSRO and AMSU-A data should be accounted for in studies such as Shangguan et al. (2019) and Bohlinger et al. (2014). These studies use both ERA5 and ERA-interim to assess long term and short term stratospheric temperature variability. In future works exploring stratospheric temperature trends, changes in temperature biases presented here need to be considered.

The temperature lidars, whilst limited to a few locations globally, have high vertical resolution measurements that have been made for nearly 30 years, making them an important and useful tool for inferring temperature biases in reanalysis datasets, which span similar time periods. A future test could see the lidar networks used to explore modifications to reanalysis datasets such as testing the experimental ERA5.1 discussed in Simmons et al. (2020).

*Data availability.* The temperature lidar data is available for public download through http://ndacc-lidar.org/index.php?id=70/Data.htm. The Microwave Limb Sounder data was available for public download at https://mls.jpl.nasa.gov/. ECMWF ERA-interim and ERA5 data are available from the ECMWF MARS archive.

*Author contributions.* GJM extracted the datasets, performed the analysis and prepared the manuscript, RW provided and processed the OHP data, ACP, RGH, IP, AH and PK provided inputs into the analysis and preparation of the manuscript, TB provided MLO and TMO data and assisted with manuscript preparation, WS provided the HOH data and assisted with manuscript preparation.

*Competing interests.* There are no competing interests

*Acknowledgements.* This work was performed during the course of the ARISE2 collaborative infrastructure design study project funded by the European Commission H2020 program (grant number 653980, www.arise-project.eu).The authors also wish to acknowledge staff at the ECMWF for their discussions as this work advanced.

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

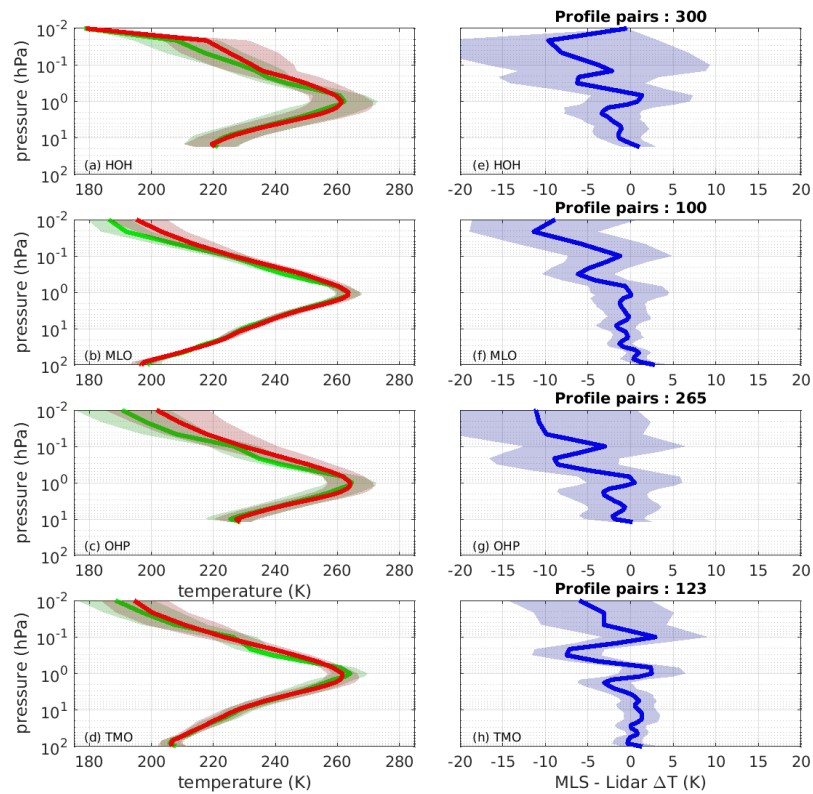

**Figure 1.** Mean profiles of both temperature from EOS MLS (green) and Rayleigh temperature lidar (red) positioned at a) Hohenpeissenburg, b) Mauna Loa, c) Observatoire de Haute-Provence and d) Table Mountain Observatory. Shading depicts 1 standard deviation in the mean temperature. The vertical profiles of the mean of the differences between the lidar and MLS for the lidar shown in a-b is shown in c-d respectively, shading shows 1 standard deviation within the mean difference.

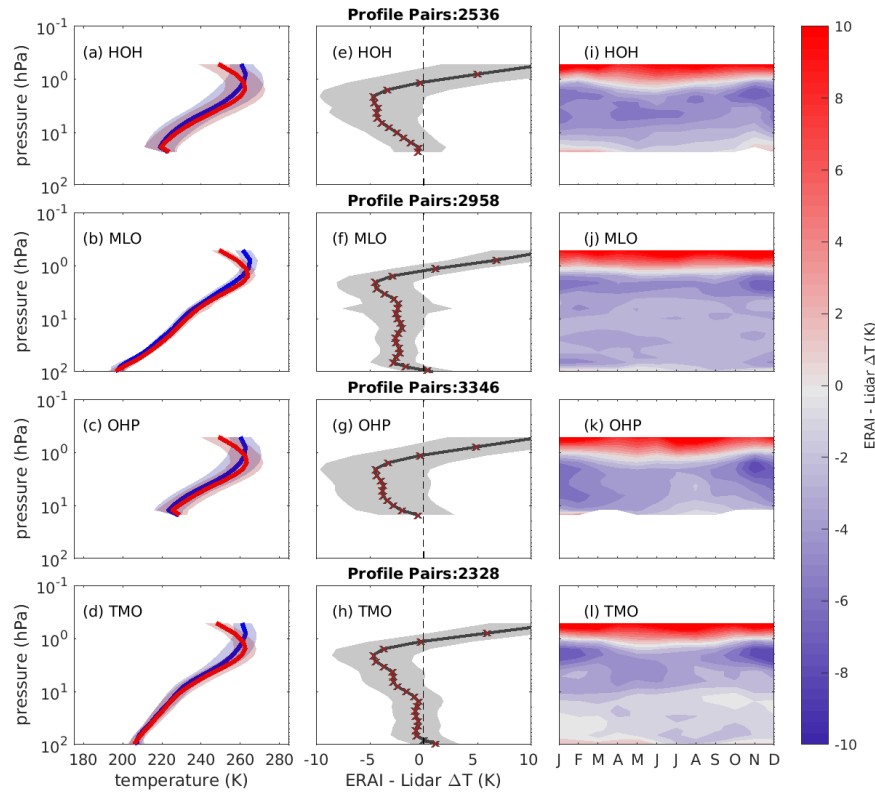

**Figure 2.** Mean profiles of matched temperatures from ERA-interim (blue) and Rayleigh temperature lidar (red) positioned at a) Hohenpeissenburg, b) Mauna Loa, c) Observatoire de Haute-Provence and d) Table Mountain Observatory between 1990 and 2017. Shading depicts 1 standard deviation in the mean temperature. The vertical profiles of the mean of the differences between ERA-interim and each lidar shown in a-d are shown in e-h respectively; shading shows 1 standard deviation of mean difference. Crosses are red where the mean of the differences were different from zero at the 95% significance level. Panels i-l show the temperature differences as a function of month and pressure level for each of the lidars shown in panels a-d

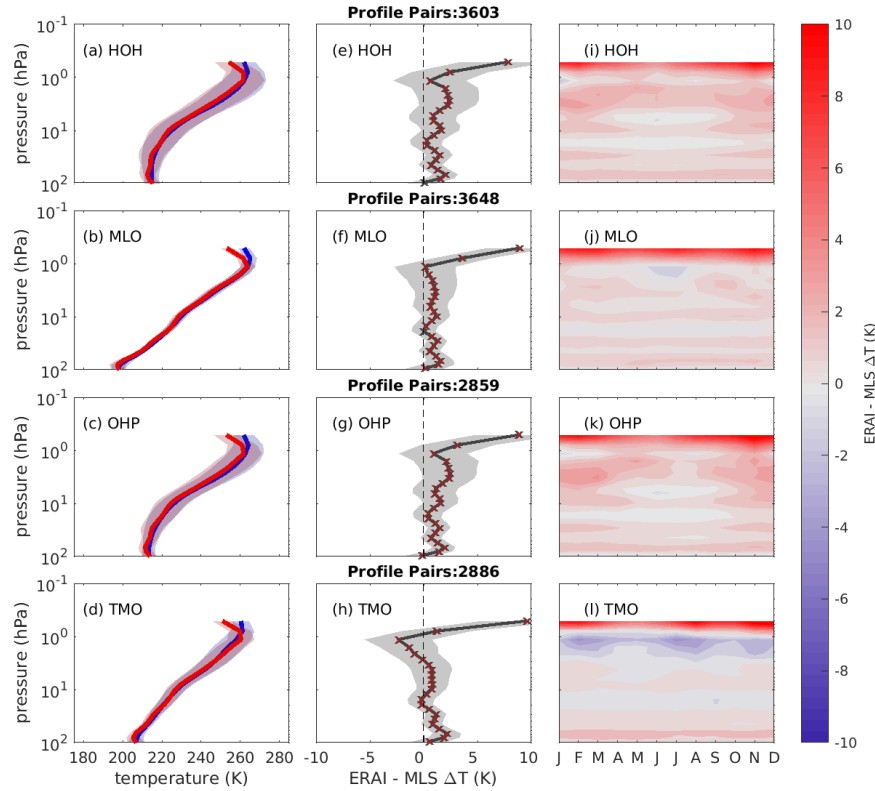

**Figure 3.** Mean profiles of matched temperatures from ERA-interim (blue) and EOS MLS passes (red) at a) Hohenpeissenburg, b) Mauna Loa, c) Observatoire de Haute-Provence and d) Table Mountain Observatory between 1990 and 2017. Shading depicts 1 standard deviation in the mean temperature. The vertical profiles of the mean of the differences between ERA-interim and EOS MLS shown in a-d are shown in e-h respectively; shading shows 1 standard deviation of mean difference. Crosses are red for model levels where the mean of the differences are different from zero at the 95% significance level. Panels i-k show the temperature differences as a function of month and pressure level for each of the lidars shown in panels a-d.

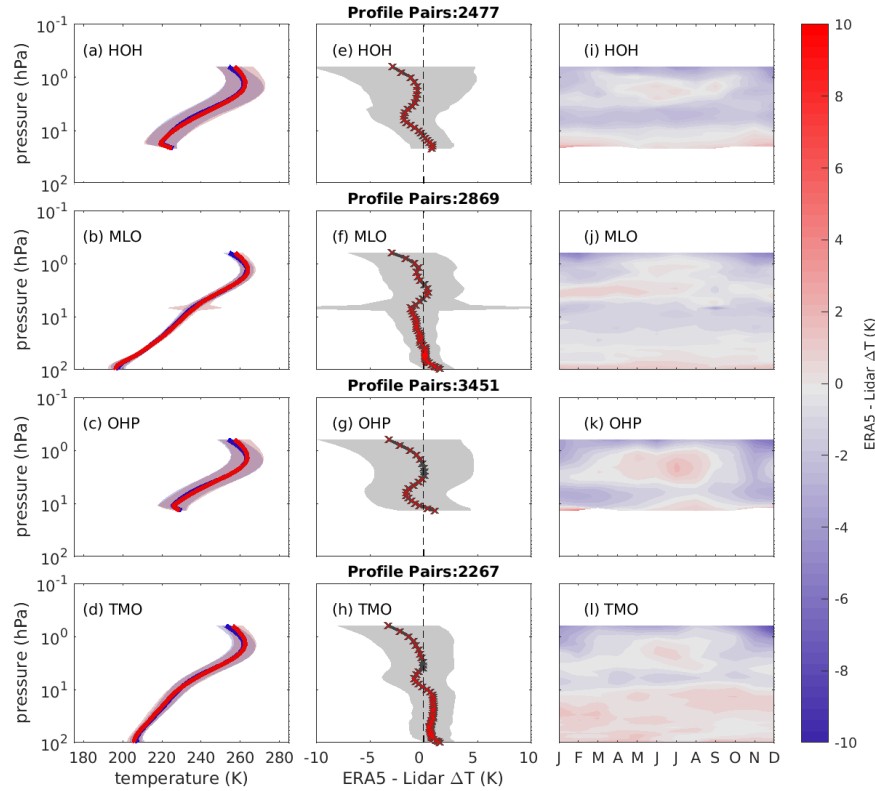

**Figure 4.** Mean profiles of matched temperatures from ERA5 (blue) and Rayleigh temperature lidar (red) positioned at a) Hohenpeissenburg, b) Mauna Loa, c) Observatoire de Haute-Provence and d) Table Mountain Observatory between 1990 and 2017. Shading depicts 1 standard deviation in the mean temperature. The vertical profiles of the mean of the differences between ERA5 and each lidar shown in a-d are shown in e-h respectively; shading shows 1 standard deviation of mean difference. Crosses are red for model levels where the mean of the differences are different from zero at the 95% significance level. Panels i-l show the temperature differences as a function of month and pressure level for each of the lidars shown in panels a-d.

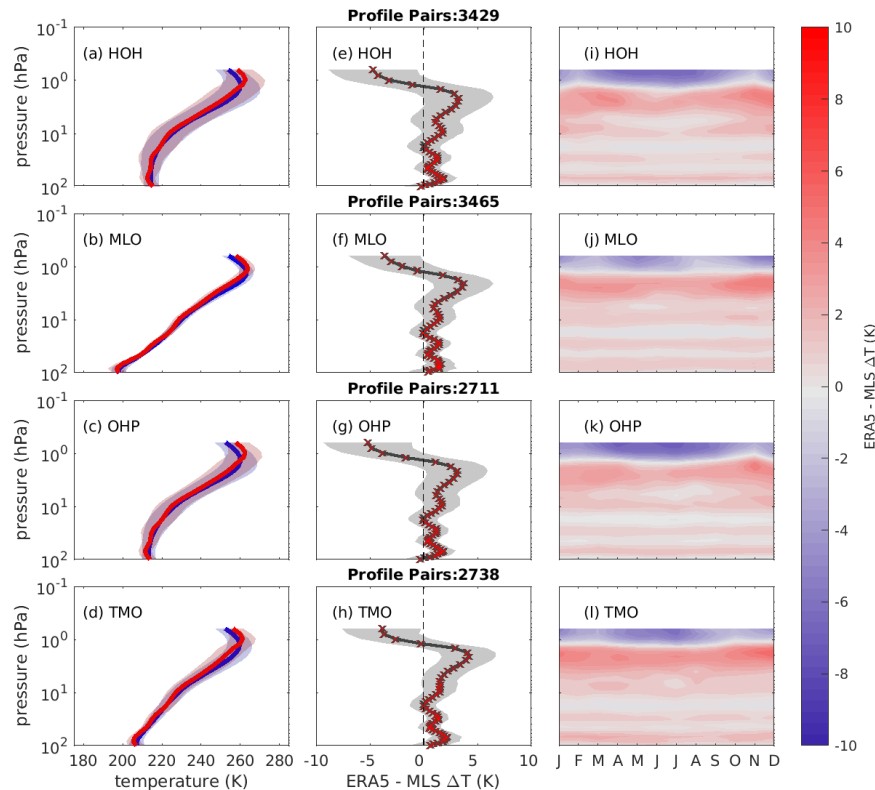

**Figure 5.** Mean profiles of matched temperatures from ERA5 (blue) and EOS MLS passes (red)at a) Hohenpeissenburg, b) Mauna Loa, c) Observatoire de Haute-Provence and d) Table Mountain Observatory between 1990 and 2017. Shading depicts 1 standard deviation in the mean temperature. The vertical profiles of the mean of the differences between ERA5 and EOS MLS are shown in a-d are shown in e-h respectively; shading shows 1 standard deviation of mean difference. Crosses are red for model levels where the mean of the differences are different from zero at the 95% significance level. Panels i-l show the temperature differences as a function of month and pressure level for each of the lidars shown in panels a-d.

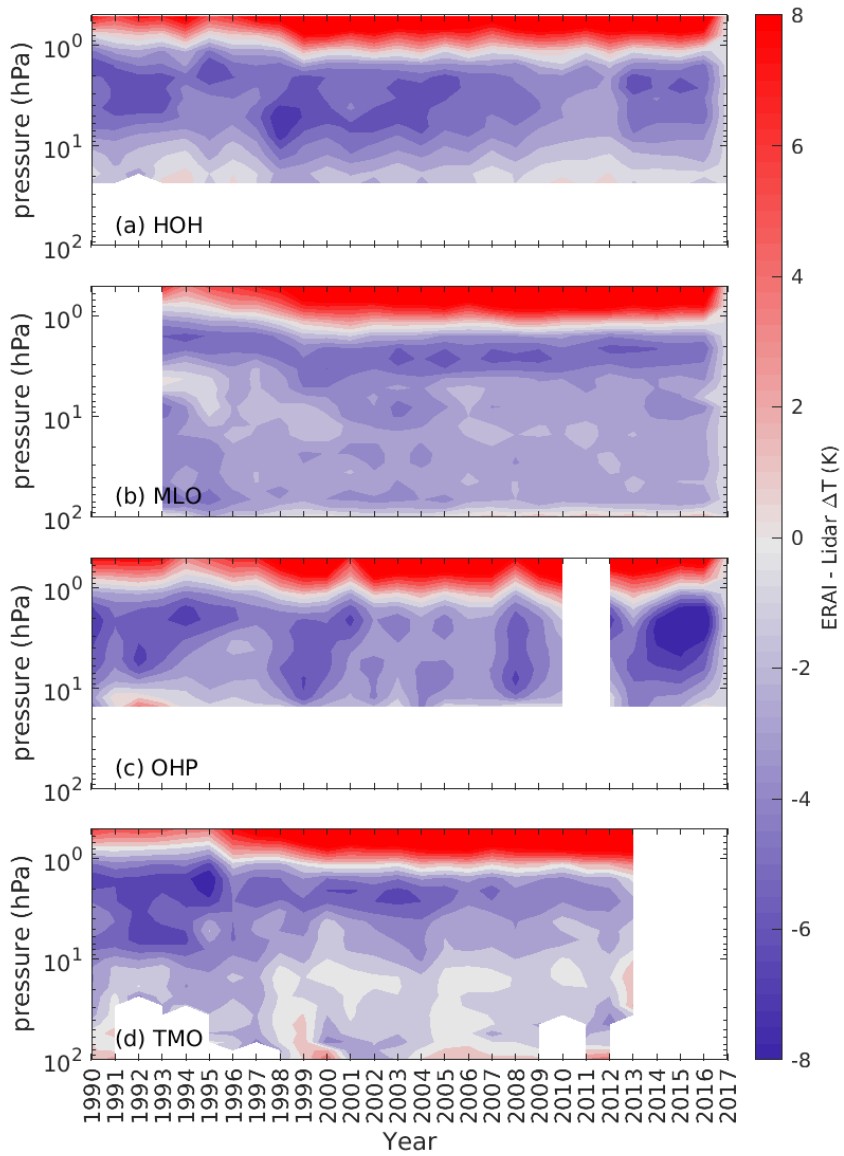

**Figure 6.** Annual Temperature bias between ERA-interim and temperature lidars at (a) Hohenpeissenburg, (b) Mauna Loa, (c) Observatoire de Haute-Provence and (d) the Table Mountain Observatory plotted as function of year and height between 1990 and 2017. Data gaps are given by white blocks.

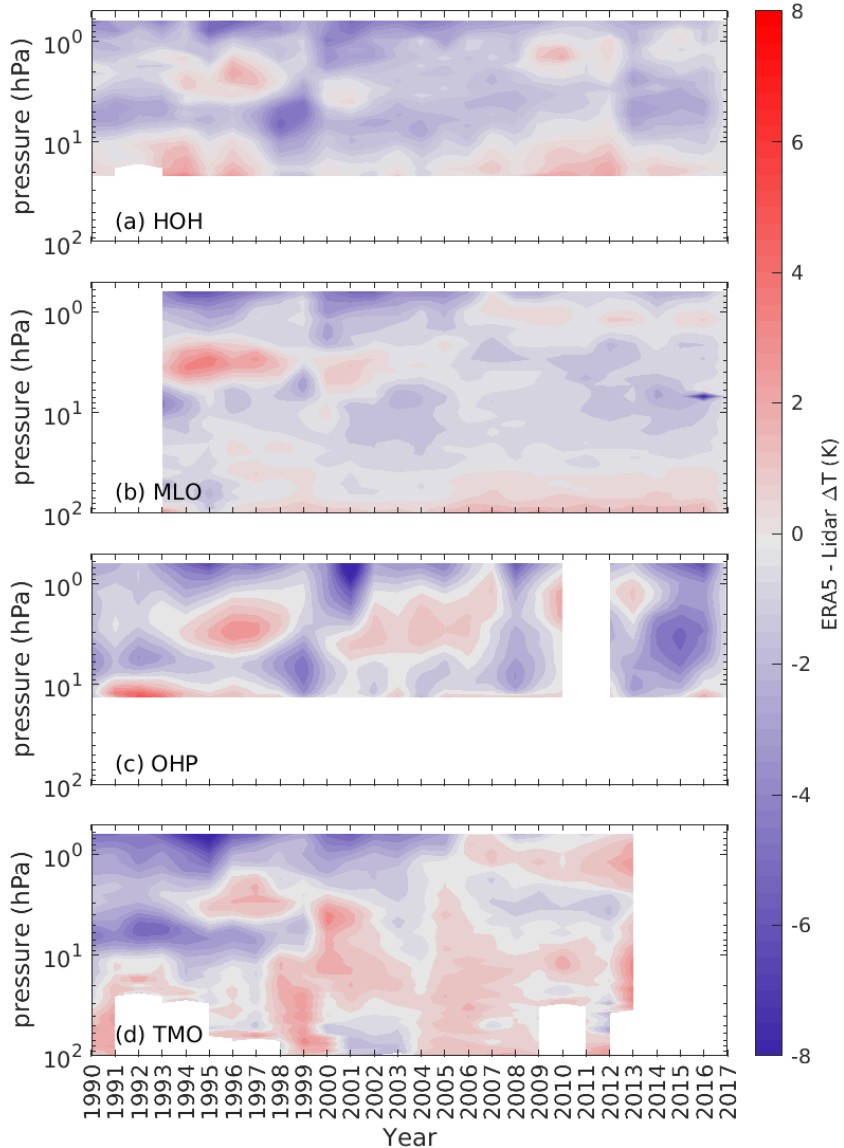

**Figure 7.** Annual temperature bias between ERA5 and temperature lidars at (a) Hohenpeissenburg, (b) Mauna Loa, (c) Observatoire de Haute-Provence and (d) the Table Mountain Observatory plotted as function of year and height between 1990 and 2017. Data gaps are given by white blocks

.

**Table 1.** Table summarising the geo-spatial and technical information of the 6 NDACC lidars used in this study

| Lidar | Lat.$^o$ | Lon.$^o$ | Period studied[1] | Wavelength (nm) | Range gate $\Delta z$ (m) |
|---|---|---|---|---|---|
| Hohenpeissenburg (HOH), Germany | 47.8 N | 11.0 E | 1987-2017 | 353 | 300 |
| Mauna Loa (MLO), Hawaii | 19.8 N | 155.7 W | 1993 - 2017 | 353 / 355[2] | 300 |
| Observatoire de Haute-Provence (OHP), France | 43.9 N | 5.7 E | 1990-2016 | 532 | 1000 |
| Table Mountain Observatory (TMO), California., US | 34.5 N | 117.7 W | 1988-2014 | 353 / 355[2] | 300 |

[1] based on data availability, [2] post 2001 upgrade.