# Peer review of "Using a network of temperature lidars to identify temperature biases in the upper stratosphere in ECMWF reanalyses"

_Atmospheric Chemistry and Physics, 2020_

## Referee Comment (RC1) · Anonymous Referee #1 · 21 Nov 2020

General Comments

Archived stratospheric temperature measurement data were compared with the ECMWF reanalysis data (ERA). The more recent ERA-5 version is shown to have better agreement with the measurements than the ERA-Interim version. This is useful for the ongoing development of the ERA reanalysis, but there are no specific scientific questions addressed in the manuscript.

I understand that the perspective of the paper is centred on comparing ERA with independent measurements. There is also a difference between the lidar and MLS measurements that is implicit in the results, as the lidar and MLS measurements have

different offsets from ERA. Why not add a figure with a comparison between the lidar and the corresponding MLS measurements at each location, and provide some brief consideration of any significant differences? As the work has already been done to match the measurements in time and location, I assume the comparison of lidar and MLS measurements would not require a lot of additional work. It would be a unique contribution to have a comparison between lidar and MLS within the context of ERA. The paper would be more significant and interesting, at least to the lidar and MLS communities.

Specific Comments

A) In describing the various sources of uncertainty in the measurements, there should be stronger distinction between random error and bias. The difference is important when assessing the averaged differences between the measurements and the model. For example, the uncertainty associated with the correction for non-linear photon counting detection in lidar measurements is a bias, and this is quite different from the random uncertainty associated with the statistics of photon counting detection. The random uncertainty is reduced by averaging, but the bias remains.

B) The description of "background noise" is not very well defined since "noise" is usually associated with random uncertainty. For example, the term "background noise extraction" at page 3, line 25. The constant background signal due to ambient light was subtracted. The "noise" in the background due to the photon counting statistics (Poisson distribution with variance equal to number of counts) cannot be extracted. It remained after the constant background was subtracted. Another example at page 4, line 10: "background noise correction uncertainty". Background is subtracted, but random noise is not corrected.

C) In this reviewer's opinion, phrases with the words "could be", "likely", "do hint at", "may be" etc. are not appropriate for a scientific publication.

D) The conclusion is not substantial. E.g. page 10, line 4: " . . . should be accounted

for." How should it be accounted for and what are the scientific implications?

Corrections in addition to the recent edit of the manuscript

Page 3, line 21: . . . laser light scattering FROM molecules and particles.

Page 5, line 4: One such "travelling standard"

Figure 4 caption: ERA-5 rather than ERA-interim.

---

## Referee Comment (RC2) · Anonymous Referee #2 · 30 Nov 2020

General Comments

This paper details temperature biases of two ECMWF datasets (ERA-interim and ERA-5) with respect to two independent observational datasets (ground-based lidar and satellite-based MLS). Biases are calculated from long-term comparisons of over 20 years for four NH lidars and ~14 years for MLS, focusing on the "wintertime" period (October-March). In addition, lidar comparisons were made for Pre-2000, 2000-2007, and Post-2007 to assess impacts of inclusion of different satellite sources. The results show a significant cold bias with respect to lidar in the upper stratosphere of ~3-4 K in ERA-interim and a warm bias above 3hPa, both of which are reduced in ERA-

5. The paper is generally well-written, and the conclusions are sound. What is less satisfactory, at least to this reviewer, is the lack of analysis of how biases may vary with time over the reanalysis. The brief look at the three time periods is helpful, but it seems that there are interesting results to be "mined" in the datasets beyond the long-term means.

Specific Comments

1. The title may be a bit misleading. It says, "Using a global network of temperature lidars", but only a very limited set of lidars (4) is actually used. Can this be considered "global"?

2. It would be very interesting to include time series of differences, in order to have closer look at the data. At a minimum, you could do plots at a few different pressure levels with a data-point for each year of the time series used (monthly means could also work, as in Figure 3 of Simmons et al. (2020)). This would make the paper more scientifically interesting, for example to see whether there are trends in the biases, or whether certain years showed much larger biases than others.

3. I am wondering why the time-averages are limited to the "winter" months (although technically this includes late fall to early spring). No explanation seems to be given why these months were chosen, except that "stratospheric variability increases in these months and we examine whether this variability is present within the reanalyses". Were similar biases not seen in the summer, or is there a reason not to look at the summer months?

4. It would be also interesting to analyze the seasonal variation in the biases by lumping together all years for each month. But maybe that is beyond the scope of this paper.

5. There is not any discussion about how background error covariances may be influencing the results, as in Simmons et al. (2020). Maybe this is beyond the scope of this paper, but some understanding of how the data assimilation assumptions are

impacting the results could be useful. Particularly, could different assumptions on ERA-interim and ERA-5 help explain some of the differences, or are they mainly due to the different datasets assimilated?

6. Page 4: The discussions of errors on this page is a bit confusing. It is difficult to tell whether the quoted errors are systematic, random, or some combination (e.g., RMS). The discussion of MLS errors did a better job of specifying errors as precision and biases.

7. One minor question about the GPSRO observations. Although you quote that they are available up to 40 km, the impact seems to reduce the bias from 3 hPa to 0.5 hPa. Much of this pressure range is generally above 40 km. Is the improvement above 40 km due to the improvements lower down impacting the upper levels via hydrostatic balance? Or do the analysis increments have correlation lengths that extend above the observations? Some comment on this would be helpful.

8. Page 5, Line 24ff: Could you also include the vertical range of these two reanalyses, both in model top and in what pressure range are the data made available?

9. Page 5, Line 33: May want to mention which of the "more measurements" may have an important impact on this study. Also, what observations are you referring to with "improved bias correction techniques".

10. Page 5, Line 34: What do you mean by "climate forcings"?

11. Page 6, Line 19: Out of curiosity, it would be helpful to know how many profiles went into the means for each site, both for lidars and for MLS. Could these numbers be included in each panel, or in the table?

12. Page 6, Line 26: Regarding the warm bias in ERA-Interim above 3 hPa, it would be useful to know whether the mean ERA-Interim profiles ever "turn over" to indicate the stratopause. That is clearly seen in the lidar, and is also seen in both ERA-5 and MLS. But maybe poor resolution of ERA-Interim at these levels makes it unable to capture

the stratopause very well. If that is the case, then the large warm bias in ERA-Interim at high altitudes could simply be explained.

13. Page 7, Line 33: You mention "uncertainties in the bias" as related to the increased width of the standard deviation. Maybe I don't understand how exactly you're calculating the standard deviation. Could you please provide some more detail? Is the smaller standard deviation for the lidar due to the longer time series used, which reduces the standard deviation of the mean?

14. Page 7, Line 36: Is the oscillation with height in the MLS data associated with specific MLS retrieval levels?

15. Page 8, Line 34: You say the profiles aren't affected significantly, but there are clearly some noticeable changes.

16. The vertical profiles of differences are a bit confusing, particularly with respect to the placement of the model levels. I'm assuming there is a solid black line with red dots overlaid only at levels where the differences are significant. Are there also black dots at levels were differences are not significant? It is difficult to distinguish these two cases from the plots. Maybe removing the black line would help, just showing dots at each level, or making the black line thin so the black dots are more easily visible. As it is, it looks like there are large regions of insignificant differences in some of the plots, where that probably isn't the case.

17. It was difficult to distinguish the colors in Figures 5 and 6, particularly which of the shaded regions corresponds to green and which to cyan. The overlapping of the shadings may be part of the confusion. Maybe an additional key for the shading would be helpful. Alternately, one could use thin lines with the same colors as the thick lines to indicate the boundaries.

Technical Corrections

1. Page 2, Line 3: May want to define middle atmosphere

2. Page 2, Line 8: typo "ReAnalaysis", also "Center" should be "Center" for NCAR.

3. Page 2, Line 12: may want to capitalize Earth to be consistent with a later reference.

4. Page 2, Line 17: comma after "data" should be a period.

5. Page 2, Line 19: May want to define upper atmosphere to distinguish from middle atmosphere.

6. Page 6, Line 16: Remove hyphen in "co-ordinate"

7. Page 6, Line 36: capitalize "Mountain Observatory"

8. Page 7, Line 8: May want to reword "Simmons et al. (2020)'s"

9. Page 7, line 24: Capitalize "Mountain"

10. Page 7, Line 31: Table Mountain Observatory is referenced as panel (g), but it is panel (h) in Figure 4.

11. Page 8, Line 34: "does not seem to [be] affected"

12. Page 9, Line 9: Hyphen after "differences" seems misplaced.

13. Page 12: Line 12: Capitalize "Observatory"

14. Figure 6. Should "table mountain" be capitalized?

15. Page 11, Line 32: Need more details on this reference (e.g., URL).

16. Page 11, Line 35: Missing source of this reference.

17. Page 12, Line 23: The Kuo reference has no journal indicated.

---

## Author Comment (AC1) · 14 Jan 2021

Response to Anonymous reviewer 1:

We thank the reviewer for their comments, our responses and proposed amendments to the manuscript are below marked with AR

General Comments: Archived stratospheric temperature measurement data were compared with the ECMWF reanalysis data (ERA). The more recent ERA-5 version is shown to have better agreement with the measurements than the ERA-Interim version. This is useful for the ongoing development of the ERA reanalysis, but there are

[Figure]

no specific scientific questions addressed in the manuscript. I understand that the perspective of the paper is centred on comparing ERA with independent measurements. There is also a difference between the lidar and MLS measurements that is implicit in the results, as the lidar and MLS measurements have different offsets from ERA. Why not add a figure with a comparison between the lidar and the corresponding MLS measurements at each location, and provide some brief consideration of any significant differences? As the work has already been done to match the measurements in time and location, I assume the comparison of lidar and MLS measurements would not require a lot of additional work. It would be a unique contribution to have a comparison between lidar and MLS within the context of ERA. The paper would be more significant and interesting, at least to the lidar and MLS communities.

AR: A Lidar-MLS comparison plot can easily be added and described in the revised manuscript. In response to the reviewers comment "there are no specific scientific questions addressed". The questions addressed in this paper are: What is the temperature bias inferred in the ERA-interim and ERA-5 reanalysis data by an independent measurement technology; ground based temperature lidar. An additional question is how does this bias change in ERA5 over time with the addition of other observations streams into the reanalysis. The introduction and conclusions will be rephrased to emphasise this more.

Specific Comments

A) In describing the various sources of uncertainty in the measurements, there should be stronger distinction between random error and bias. The difference is important when assessing the averaged differences between the measurements and the model. For example, the uncertainty associated with the correction for non-linear photon counting detection in lidar measurements is a bias, and this is quite different from the random uncertainty associated with the statistics of photon counting detection. The random uncertainty is reduced by averaging, but the bias remains.

AR: We will check the manuscript for all mentions of uncertainty and ensure we are correctly referring to random error and bias, an offset between either two measurement technologies or models are referred to correctly. In response to the more specific comments regarding uncertainty surrounding the corrections in the lidars. From a metrology point anything that is not signal measured by the lidar is noise. This noise can be split into a random components in the form of photon detection, which becomes negligible when averaging, and a systematic component or bias such as level of sky light etc, which can be budgeted and corrections applied for e.g. (Leblanc et al 2016). The text in section 2 of the revised manuscript will be changed appropriately.

B) The description of "background noise" is not very well defined since "noise" is usually associated with random uncertainty. For example, the term "background noise extraction" at page 3, line 25. The constant background signal due to ambient light was subtracted. The "noise" in the background due to the photon counting statistics (Poisson distribution with variance equal to number of counts) cannot be extracted. It remained after the constant background was subtracted. Another example at page4, line 10: "background noise correction uncertainty". Background is subtracted, but random noise is not corrected.

AR: Following discussion above this will be reworded to emphasise that background noise which is systematic can be extracted and removed where random noise from photon counting cannot be corrected.

C) In this reviewer's opinion, phrases with the words "could be", "likely", "do hint at","may be" etc. are not appropriate for a scientific publication.

AR: The manuscript will undergo a thorough proofreading to replace these phrases with more appropriate ones.

D) The conclusion is not substantial. E.g. page 10, line 4: "...should be accounted for." How should it be accounted for and what are the scientific implications?

AR: Studies such as Shangguan et al (2019) and Bohlinger et al (2014) use both ERA5 and ERA-interim to assess long term and short term stratospheric temperature variability in the stratosphere. In future works exploring stratospheric temperature trends changes in temperature bias presented here in this work, will need consideration when analysing results. This will be added to the conclusions in the revised manuscript

Corrections in addition to the recent edit of the manuscript Page 3, line 21:...laser light scattering FROM molecules and particles. Page 5, line 4: One such "travelling standard" Figure 4 caption: ERA-5 rather than ERA-interim

AR: These will be amended in the revised manuscript

References: Bohlinger, P., Sinnhuber, B.M., Ruhnke, R. and Kirner, O., 2014. Radiative and dynamical contributions to past and future Arctic stratospheric temperature trends. Atmospheric Chemistry & Physics Discussions, 14(3).

Leblanc, T., Sica, R.J., van Gijsel, J.A., Haefele, A., Payen, G. and Liberti, G., 2016. Proposed standardized definitions for vertical resolution and uncertainty in the NDACC lidar ozone and temperature algorithms-Part 3: Temperature uncertainty budget. Atmospheric Measurement Techniques, 9(8).

Shangguan, M., Wang, W. and Jin, S., 2019. Variability of temperature and ozone in the upper troposphere and lower stratosphere from multi-satellite observations and reanalysis data. Atmospheric Chemistry and Physics, 19(10), pp.6659-6679.
* * *

---

## Author Comment (AC2) · 14 Jan 2021

Reviewer 2: We thank the reviewer for their comments, our responses and proposed amendments to the manuscript are below marked by AR

This paper details temperature biases of two ECMWF datasets (ERA-interim and ERA-5) with respect to two independent observational datasets (ground-based lidar and satellite-based MLS). Biases are calculated from long-term comparisons of over 20years for four NH lidars andâĹij14 years for MLS, focusing on the "wintertime" period(October-March). In addition, lidar comparisons were made for Pre-2000, 2000-2007,and Post-2007 to assess impacts of inclusion of different satellite sources. The

results show a significant cold bias with respect to lidar in the upper stratosphere ofâĹij3-4K in ERA-interim and a warm bias above 3hPa, both of which are reduced in ERA. The paper is generally well-written, and the conclusions are sound. What is less satisfactory, at least to this reviewer, is the lack of analysis of how biases may vary with time over the reanalysis. The brief look at the three time periods is helpful, but it seems that there are interesting results to be "mined" in the datasets beyond the long-term means.

Specific Comments 1. The title may be a bit misleading. It says, "Using a global network of temperature lidars", but only a very limited set of lidars (4) is actually used. Can this be considered "global"?

AR: Global will be removed from the title

2. It would be very interesting to include time series of differences, in order to have closer look at the data. At a minimum, you could do plots at a few different pressure levels with a data-point for each year of the time series used (monthly means could also work, as in Figure 3 of Simmons et al. (2020)). This would make the paper more scientifically interesting, for example to see whether there are trends in the biases, or whether certain years showed much larger biases than others.

3. I am wondering why the time-averages are limited to the "winter" months (although technically this includes late fall to early spring). No explanation seems to be given why these months were chosen, except that "stratospheric variability increases in these months and we examine whether this variability is present within the reanalyses". Were similar biases not seen in the summer, or is there a reason not to look at the summer months?

4. It would be also interesting to analyze the seasonal variation in the biases by lumping together all years for each month. But maybe that is beyond the scope of this paper.

AR: The time averages were limited to winter months, initially to see if the heightened

stratospheric variability was represented. We realise this is a shortcoming and propose, to include an additional panel in figures 1,2,3 and 4 showing a contour plot of the temperature difference by month for each station as well as recomputing the mean difference for all months. Addressing point 2 we will then create a time series of temperature bias at a selection of pressure levels as suggested by the reviewer to replace figures 5 and 6

5. There is not any discussion about how background error covariances may be influencing the results, as in Simmons et al. (2020). Maybe this is beyond the scope of this paper, but some understanding of how the data assimilation assumptions are impacting the results could be useful. Particularly, could different assumptions on ERA-interim and ERA-5 help explain some of the differences, or are they mainly due to the different datasets assimilated?

AR: Given the differences between ERA-interim (Dee et al 2011) and ERA-5 (Hersbach et al 2020), there are differences in both the vertical and horizontal resolution, model physics, data assimilation and observation streams. All of which could potentially explain and contribute to the results seen here. However, it is hard to imply from our analysis a single dominant component.

6. Page 4: The discussions of errors on this page is a bit confusing. It is difficult to tell whether the quoted errors are systematic, random, or some combination (e.g., RMS). The discussion of MLS errors did a better job of specifying errors as precision and biases. AR: We will work on the lidar section to define the systematic and random errors more clearly.

7. One minor question about the GPSRO observations. Although you quote that they are available up to 40 km, the impact seems to reduce the bias from 3 hPa to 0.5 hPa. Much of this pressure range is generally above 40 km. Is the improvement above 40km due to the improvements lower down impacting the upper levels via hydrostatic balance? Or do the analysis increments have correlation lengths that extend above the

observations? Some comment on this would be helpful.

AR: First a correction: the average assimilation height of GPS-RO is to 50 km within ERA5. In addition to this, the 50 km height is a tangent height and the bending angle is then integrated from the tangent point upwards to the model top so the introduction of GPSRO will have effects above 50 km (Healy 2008). In addition to this structure functions used in the data assimilation and the dynamics of the assimilating model used for ERA5 will influence the model levels surrounding it. This discussion will be added to the revised manuscript.

8. Page 5, Line 24: Could you also include the vertical range of these two reanalyses, both in model top and in what pressure range are the data made available? AR: ERA interim has a 60 model level range from ∼1000hPa to 0.3 hPa. ERA5 has 137 model levels and a pressure range ∼1000hPa to 0.02 hPa. However, the top 10 levels of ERA5 are not used to give similar vertical range to the comparisons with ERA-5, these values will be added to the manuscript

9. Page 5, Line 33: May want to mention which of the "more measurements" may have an important impact on this study. Also, what observations are you referring to with "improved bias correction techniques"

AR: Section 5 of Hersbach et al (2020) details the new and reprocessed data sets used in ERA-5 that were not present in ERA-interim. The revised manuscript will include a reference to this citation in section 2.3

10. Page 5, Line 34: What do you mean by "climate forcings"?

AR: Hersbach et al 2020. Section 6.1 describes CMIP5 radiative forcings. The revised manuscript will include this citation in section 2.3 as well as changing climate forcings to CMIP5 radiative forcings

11. Page 6, Line 19: Out of curiosity, it would be helpful to know how many profiles went into the means for each site, both for lidars and for MLS. Could these numbers be

included in each panel, or in the table?

AR: Profile counts will be added to table 1 in the revised manuscript.

12. Page 6, Line 26: Regarding the warm bias in ERA-Interim above 3 hPa, it would be useful to know whether the mean ERA-Interim profiles ever "turn over" to indicate the stratopause. That is clearly seen in the lidar, and is also seen in both ERA-5 and MLS.But maybe poor resolution of ERA-Interim at these levels makes it unable to capture the stratopause very well. If that is the case, then the large warm bias in ERA-Interim at high altitudes could simply be explained.

AR: Due to ERA-interim approaching the model top it would not be possible to show the stratopause overturn. We will add to this section remarks about the large warm bias being an artefact of approaching the model top.

13. Page 7, Line 33: You mention "uncertainties in the bias" as related to the increased width of the standard deviation. Maybe I don't understand how exactly you're calculating the standard deviation. Could you please provide some more detail? Is the smaller standard deviation for the lidar due to the longer time series used, which reduces the standard deviation of the mean?

AR: This will be reworded in the revised manuscript to say that the standard deviation in the temperature differences between MLS and ERA increase. This increase in standard deviation increases the uncertainty of what the bias is. Due to the large number of profiles for both datasets the profile count won't affect the standard deviation.

14. Page 7, Line 36: Is the oscillation with height in the MLS data associated with specific MLS retrieval levels?

AR: The oscillation here cannot solely be attributed to the MLS retrieval levels. Some Oscillatory effects have been seen in Wing et al 2018. Wing et al 2018 showed large oscillations in temperature difference between MLS and Lidar that spanned several retrieval levels. This may explain the larger oscillatory behaviour at the top of the

profile in figure 4. The oscillations with smaller vertical length scale with variability less than 1 K at lower altitudes falls within the quoted precision for that height range shown discussed in section 2.2. We will include this discussion in our revised manuscript

15. Page 8, Line 34: You say the profiles aren't affected significantly, but there are clearly some noticeable changes.

AR: This plot will be replaced with a temperature time series at several pressure levels for each site

16. The vertical profiles of differences are a bit confusing, particularly with respect to the placement of the model levels. I'm assuming there is a solid black line with red dots overlaid only at levels where the differences are significant. Are there also black dots at levels were differences are not significant? It is difficult to distinguish these two cases from the plots. Maybe removing the black line would help, just showing dots at each level, or making the black line thin so the black dots are more easily visible. As it is, it looks like there are large regions of insignificant differences in some of the plots, where that probably isn't the case

AR: Red dots were placed at levels where there was a significant difference. However, as the reviewer points out it is hard to distinguish the model levels where there is not a significant difference. In the revised manuscript we will change the plotting to better show the model levels and the significance of the temperature difference.

17. It was difficult to distinguish the colors in Figures 5 and 6, particularly which of the shaded regions corresponds to green and which to cyan. The overlapping of the shadings may be part of the confusion. Maybe an additional key for the shading would be helpful. Alternately, one could use thin lines with the same colors as the thick lines to indicate the boundaries.

AR: This plot will be replaced with a time series plot of the differences at several pressure levels in the revised manuscript

Technical Corrections 1. Page 2, Line 3: May want to define middle atmosphere

2. Page 2, Line 8: typo "ReAnalaysis", also "Center" should be "Center" for NCAR.

3. Page 2, Line 12: may want to capitalize Earth to be consistent with a later reference.

4. Page 2, Line 17: comma after "data" should be a period.

5. Page 2, Line 19: May want to define upper atmosphere to distinguish from middle atmosphere.

6. Page 6, Line 16: Remove hyphen in "co-ordinate"

7. Page 6, Line 36: capitalize "Mountain Observatory"

8. Page 7, Line 8: May want to reword "Simmons et al. (2020)'s"

9. Page 7, line 24: Capitalize "Mountain"

10. Page 7, Line 31: Table Mountain Observatory is referenced as panel (g), but it ispanel (h) in Figure 4.

11. Page 8, Line 34: "does not seem to [be] affected"

12. Page 9, Line 9: Hyphen after "differences" seems misplaced.

13. Page 12: Line 12: Capitalize "Observatory"

14. Figure 6. Should "table mountain" be capitalized?

15. Page 11, Line 32: Need more details on this reference (e.g., URL).

16. Page 11, Line 35: Missing source of this reference.

17. Page 12, Line 23: The Kuo reference has no journal indicated.

AR: These will be addressed in the revised manuscript

References

Dee, D.P., Uppala, S.M., Simmons, A.J., Berrisford, P., Poli, P., Kobayashi, S., Andrae, U., Balmaseda, M.A., Balsamo, G., Bauer, D.P. and Bechtold, P., 2011. The ERA‐Interim reanalysis: Configuration and performance of the data assimilation system. Quarterly Journal of the royal meteorological society, 137(656), pp.553-597.

Healy, S.B., 2008, June. Assimilation of GPS radio occultation measurements at ECMWF. In Proceedings of the GRAS SAF Workshop on Applications of GPSRO measurements, ECMWF, Reading, UK (pp. 16-18).

Hersbach, H., Bell, B., Berrisford, P., Hirahara, S., Horányi, A., Muñoz‐Sabater, J., Nicolas, J., Peubey, C., Radu, R., Schepers, D. and Simmons, A., 2020. The ERA5 global reanalysis. Quarterly Journal of the Royal Meteorological Society, 146(730), pp.1999-2049.

Robin Wing, Alain Hauchecorne, Philippe Keckhut, Sophie Godin-Beekmann, Sergey Khaykin, et al.. Lidar temperature series in the middle atmosphere as a reference data set – Part 2: Assessment of temperature observations from MLS/Aura and SABER/TIMED satellites. Atmospheric Measurement Techniques, European Geosciences Union, 2018, 11 (12), pp.6703-6717. ff10.5194/amt-11-6703-2018ff. ffinsu-01784812f

---

## Author Response (AR1)

**Response to Anonymous reviewer 1:**

We thank the reviewer for their comments, our responses to the comments are below. Changes to the manuscript have been marked in the tracked change version

*General Comments:*

*Archived stratospheric temperature measurement data were compared with the ECMWF reanalysis data (ERA). The more recent ERA-5 version is shown to have better agreement with the measurements than the ERA-Interim version. This is useful for the ongoing development of the ERA reanalysis, but there are no specific scientific questions addressed in the manuscript.*

*I understand that the perspective of the paper is centred on comparing ERA with independent measurements. There is also a difference between the lidar and MLS measurements that is implicit in the results, as the lidar and MLS measurements have different offsets from ERA. Why not add a figure with a comparison between the lidar and the corresponding MLS measurements at each location, and provide some brief consideration of any significant differences? As the work has already been done to match the measurements in time and location, I assume the comparison of lidar and MLS measurements would not require a lot of additional work. It would be a unique contribution to have a comparison between lidar and MLS within the context of ERA. The paper would be more significant and interesting, at least to the lidar and MLS communities.*

A Lidar-MLS comparison plot has been added and described in the text in the revised manuscript. In response to the reviewer's comment "there are no specific scientific questions addressed". The questions addressed in this paper are: What is the temperature bias inferred in the ERA-interim and ERA-5 reanalysis data by an independent measurement technology; ground based temperature lidar. An additional question is how does this bias change in ERA5 over time with the addition of other observations streams into the reanalysis. The introduction and conclusions will be rephrased to emphasise this more.

*Specific Comments*

A) *In describing the various sources of uncertainty in the measurements, there should be stronger distinction between random error and bias. The difference is important when assessing the averaged differences between the measurements and the model. For example, the uncertainty associated with the correction for non-linear photon counting detection in lidar measurements is a bias, and this is quite different from the random uncertainty associated with the statistics of photon counting detection. The random uncertainty is reduced by averaging, but the bias remains.*

We have checked the manuscript for all mentions of uncertainty and ensure we are correctly referring to random error and bias, an offset between either two measurement technologies or models are referred to correctly.

In response to the more specific comments regarding uncertainty surrounding the corrections in the lidars. From a metrology point anything that is not signal measured by the lidar is noise. This noise can be split into a random component in the form of photon detection, which becomes negligible when averaging, and a systematic component or bias such as level of sky light etc, which can be budgeted and corrections applied for e.g. (Leblanc et al 2016). The text in section 2 of the revised manuscript has been changed appropriately.

B) *The description of "background noise" is not very well defined since "noise" is usually associated with random uncertainty. For example, the term "background noise extraction" at page 3, line 25. The constant background signal due to ambient light was subtracted. The "noise" in the background due*

*to the photon counting statistics (Poisson distribution with variance equal to number of counts) cannot be extracted. It remained after the constant background was subtracted. Another example at page4, line 10: "background noise correction uncertainty". Background is subtracted, but random noise is not corrected.*

Following discussion above this has been reworded to emphasise that background noise which is systematic can be estimated (extracted) and removed where random noise from photon counting, which cannot be corrected, can only be reduced in reducing vertical or time resolution through averaging.

C)   *In this reviewer's opinion, phrases with the words "could be", "likely", "do hint at", "may be" etc. are not appropriate for a scientific publication.*

The manuscript has undergone a thorough proofreading to replace these phrases with more appropriate ones

D)   *The conclusion is not substantial. E.g. page 10, line 4: "...should be accounted for." How should it be accounted for and what are the scientific implications?*

Studies such as Shangguan et al (2019) and Bohlinger et al (2014) use both ERA5 and ERA-interim to assess long-term and short-term stratospheric temperature variability in the stratosphere. In future works exploring stratospheric temperature trends changes in temperature bias presented here in this work, will need consideration when analysing results. This has been added to the conclusions in the revised manuscript

*Corrections in addition to the recent edit of the manuscript*

*Page 3, line 21:...laser light scattering FROM molecules and particles.*

*Page 5, line 4: One such "travelling standard"*

*Figure 4 caption: ERA-5 rather than ERA-interim*

These have been amended in the revised manuscript

References:

Bohlinger, P., Sinnhuber, B.M., Ruhnke, R. and Kirner, O., 2014. Radiative and dynamical contributions to past and future Arctic stratospheric temperature trends. *Atmospheric Chemistry & Physics Discussions*, *14*(3).

Leblanc, T., Sica, R.J., van Gijsel, J.A., Haefele, A., Payen, G. and Liberti, G., 2016. Proposed standardized definitions for vertical resolution and uncertainty in the NDACC lidar ozone and temperature algorithms-Part 3: Temperature uncertainty budget. *Atmospheric Measurement Techniques*, *9*(8).

Shangguan, M., Wang, W. and Jin, S., 2019. Variability of temperature and ozone in the upper troposphere and lower stratosphere from multi-satellite observations and reanalysis data. *Atmospheric Chemistry and Physics*, *19*(10), pp.6659-6679.

**Reviewer 2:**

We thank the reviewer for their comments, our responses to the comments are below. Changes to the manuscript have been marked in the tracked change version

*This paper details temperature biases of two ECMWF datasets (ERA-interim and ERA-5) with respect to two independent observational datasets (ground-based lidar and satellite-based MLS). Biases are calculated from long-term comparisons of over 20years for four NH lidars and ~14 years for MLS, focusing on the "wintertime" period(October-March). In addition, lidar comparisons were made for Pre-2000, 2000-2007,and Post-2007 to assess impacts of inclusion of different satellite sources. The results show a significant cold bias with respect to lidar in the upper stratosphere of ~3-4K in ERA-interim and a warm bias above 3hPa, both of which are reduced in ERA. The paper is generally well-written, and the conclusions are sound. What is less satisfactory, at least to this reviewer, is the lack of analysis of how biases may vary with time over the reanalysis. The brief look at the three time periods is helpful, but it seems that there are interesting results to be "mined" in the datasets beyond the long-term means.*

*Specific Comments*

1. *The title may be a bit misleading. It says, "Using a global network of temperature lidars", but only a very limited set of lidars (4) is actually used. Can this be considered "global"?*

Global has been removed from the title

2. *It would be very interesting to include time series of differences, in order to have closer look at the data. At a minimum, you could do plots at a few different pressure levels with a data-point for each year of the time series used (monthly means could also work, as in Figure 3 of Simmons et al. (2020)). This would make the paper more scientifically interesting, for example to see whether there are trends in the biases, or whether certain years showed much larger biases than others.*

3. *I am wondering why the time-averages are limited to the "winter" months (although technically this includes late fall to early spring). No explanation seems to be given why these months were chosen, except that "stratospheric variability increases in these months and we examine whether this variability is present within the reanalyses". Were similar biases not seen in the summer, or is there a reason not to look at the summer months?*

4. *It would be also interesting to analyze the seasonal variation in the biases by lumping together all years for each month. But maybe that is beyond the scope of this paper.*

Comparisons are now undertaken for all months of the year. Figures 1-4 (now 2-5) have a seasonal composite panel plotted showing the temperature bias by month of year. Figures 5-6 (now 6-7) have been replotted to show the yearly temperature differences by year and height. The discussion around the plots has been extended and modified where necessary

5. *There is not any discussion about how background error covariances may be influencing the results, as in Simmons et al. (2020). Maybe this is beyond the scope of this paper, but some understanding of how the data assimilation assumptions are impacting the results could be useful. Particularly, could different assumptions on ERA-interim and ERA-5 help explain some of the differences, or are they mainly due to the different datasets assimilated?*

Given the differences between ERA-interim (Dee et al 2011) and ERA-5 (Hersbach et al 2020), there are differences in both the vertical and horizontal resolution, model physics, data assimilation and observation streams. All of which could potentially explain and contribute to the results seen here.

However, it is hard to imply from our analysis a single dominant component. We have highlighted the data-assimilation systems in both ERA-interim and ERA-5 in the text and references to the papers that explain this in more detail.

6. *Page 4: The discussions of errors on this page is a bit confusing. It is difficult to tell whether the quoted errors are systematic, random, or some combination (e.g., RMS). The discussion of MLS errors did a better job of specifying errors as precision and biases.*

We have re-worded this section to make sure references to bias, precision, systematic error and random error are clearer and more defined

7. *One minor question about the GPSRO observations. Although you quote that they are available up to 40 km, the impact seems to reduce the bias from 3 hPa to 0.5 hPa. Much of this pressure range is generally above 40 km. Is the improvement above 40km due to the improvements lower down impacting the upper levels via hydrostatic balance? Or do the analysis increments have correlation lengths that extend above the observations? Some comment on this would be helpful.*

First a correction: the average assimilation height of GPS-RO is to 50 km within ERA5. In addition to this, the 50 km height is a tangent height, and the bending angle is then integrated from the tangent point upwards to the model top so the introduction of GPSRO will have effects above 50 km (Healy 2008). In addition to this structure functions used in the data assimilation and the dynamics of the assimilating model used for ERA5 will influence the model levels surrounding it. This discussion has been added to the revised manuscript.

8. *Page 5, Line 24: Could you also include the vertical range of these two reanalyses, both in model top and in what pressure range are the data made available?*

ERA interim has a 60 model level range from ~1000hPa to 0.3 hPa. ERA5 has 137 model levels and a pressure range ~1000hPa to 0.02 hPa. However, the top 10 levels of ERA5 are not used to give similar vertical range to the comparisons with ERA-5, these values have been added to the manuscript

9. *Page 5, Line 33: May want to mention which of the "more measurements" may have an important impact on this study. Also, what observations are you referring to with "improved bias correction techniques"*

Section 5 of Hersbach et al (2020) details the new and reprocessed data sets used in ERA-5 that were not present in ERA-interim. The revised manuscript will include a reference to this citation in section 2.3

10. *Page 5, Line 34: What do you mean by "climate forcings"?*

Hersbach et al 2020. Section 6.1 describes CMIP5 radiative forcing. The revised manuscript now include this citation in section 2.3 as well as changing climate forcing to CMIP5 radiative forcings.

11. *Page 6, Line 19: Out of curiosity, it would be helpful to know how many profiles went into the means for each site, both for lidars and for MLS. Could these numbers be included in each panel, or in the table?*

Matched profile counts have been added to figures 1-4 (now 2-5)

12. *Page 6, Line 26: Regarding the warm bias in ERA-Interim above 3 hPa, it would be useful to know whether the mean ERA-Interim profiles ever "turn over" to indicate the stratopause. That is clearly*

*seen in the lidar, and is also seen in both ERA-5 and MLS.But maybe poor resolution of ERA-Interim at these levels makes it unable to capture the stratopause very well. If that is the case, then the large warm bias in ERA-Interim at high altitudes could simply be explained.*

Due to ERA-interim approaching the model top it would not be possible to show the stratopause overturn. We have added to the discussion section about the large warm bias being an artefact of approaching the model top.

*13. Page 7, Line 33: You mention "uncertainties in the bias" as related to the increased width of the standard deviation. Maybe I don't understand how exactly you're calculating the standard deviation. Could you please provide some more detail? Is the smaller standard deviation for the lidar due to the longer time series used, which reduces the standard deviation of the mean?*

This comment has been retracted. However, a better description of how the grey shading representing the standard deviation of the matched temperature differences has been given.

*14. Page 7, Line 36: Is the oscillation with height in the MLS data associated with specific MLS retrieval levels?*

The oscillation here cannot solely be attributed to the MLS retrieval levels. Some Oscillatory effects have been seen in Wing et al 2018. Wing et al 2018 showed large oscillations in temperature difference between MLS and Lidar that spanned several retrieval levels. This may explain the larger oscillatory behaviour at the top of the profile in figure 4. The oscillations with smaller vertical length scale with variability less than 1 K at lower altitudes falls within the quoted precision for that height range shown discussed in section 2.2. The manuscript has been amended to state this

*15. Page 8, Line 34: You say the profiles aren't affected significantly, but there are clearly some noticeable changes.*

This plot has been replaced with a time height plot of temperature difference for each site.

*16. The vertical profiles of differences are a bit confusing, particularly with respect to the placement of the model levels. I'm assuming there is a solid black line with red dots overlaid only at levels where the differences are significant. Are there also black dots at levels were differences are not significant? It is difficult to distinguish these two cases from the plots. Maybe removing the black line would help, just showing dots at each level, or making the black line thin so the black dots are more easily visible. As itis, it looks like there are large regions of insignificant differences in some of the plots, where that probably isn't the case*

We have replaced the dots with crosses to show the model level, red crosses show a significant difference, black crosses show otherwise. The text and captions have been amended appropriately.

*17. It was difficult to distinguish the colors in Figures 5 and 6, particularly which of the shaded regions corresponds to green and which to cyan. The overlapping of theshadings may be part of the confusion. Maybe an additional key for the shading would be helpful. Alternately, one could use thin lines with the same colors as the thick lines to indicate the boundaries.*

These plots have been replaced with a time height plot of temperature difference for each site.

*Technical Corrections*

1. *Page 2, Line 3: May want to define middle atmosphere*

   This has been changed to: The middle atmosphere, spanning 10-80~km in altitude and contains the stratosphere and mesosphere,..

2. *Page 2, Line 8: typo "ReAnalaysis", also "Center" should be "Center" for NCAR.*
3. *Page 2, Line 12: may want to capitalize Earth to be consistent with a later reference.*
4. *Page 2, Line 17: comma after "data" should be a period.*
5. *Page 2, Line 19: May want to define upper atmosphere to distinguish from middleatmosphere.*
6. *Page 6, Line 16: Remove hyphen in "co-ordinate"*
7. *Page 6, Line 36: capitalize "Mountain Observatory"*
8. *Page 7, Line 8: May want to reword "Simmons et al. (2020)'s"*
9. *Page 7, line 24: Capitalize "Mountain"*
10. *Page 7, Line 31: Table Mountain Observatory is referenced as panel (g), but it ispanel (h) in Figure 4.*
11. *Page 8, Line 34: "does not seem to [be] affected"*
12. *Page 9, Line 9: Hyphen after "differences" seems misplaced.*
13. *Page 12: Line 12: Capitalize "Observatory"*
14. *Figure 6. Should "table mountain" be capitalized?*
15. *Page 11, Line 32: Need more details on this reference (e.g., URL).*
16. *Page 11, Line 35: Missing source of this reference.*
17. *Page 12, Line 23: The Kuo reference has no journal indicated.*

These have all been addressed in the revised manuscript

References

Dee, D.P., Uppala, S.M., Simmons, A.J., Berrisford, P., Poli, P., Kobayashi, S., Andrae, U., Balmaseda, M.A., Balsamo, G., Bauer, D.P. and Bechtold, P., 2011. The ERA-Interim reanalysis: Configuration and performance of the data assimilation system. *Quarterly Journal of the royal meteorological society*, *137*(656), pp.553-597.

Healy, S.B., 2008, June. Assimilation of GPS radio occultation measurements at ECMWF. In Proceedings of the GRAS SAF Workshop on Applications of GPSRO measurements, ECMWF, Reading, UK (pp. 16-18).

Hersbach, H., Bell, B., Berrisford, P., Hirahara, S., Horányi, A., Muñoz-Sabater, J., Nicolas, J., Peubey, C., Radu, R., Schepers, D. and Simmons, A., 2020. The ERA5 global reanalysis. *Quarterly Journal of the Royal Meteorological Society*, *146*(730), pp.1999-2049.

*Robin Wing, Alain Hauchecorne, Philippe Keckhut, Sophie Godin-Beekmann, Sergey Khaykin, et al.. Lidar temperature series in the middle atmosphere as a reference data set – Part 2: Assessment of temperature observations from MLS/Aura and SABER/TIMED satellites. Atmospheric Measurement Techniques, European Geosciences Union, 2018, 11 (12), pp.6703-6717. ff10.5194/amt-11-6703-2018ff. ffinsu-01784812f*

---

## Editor Decision (ED1)

Editor's comments on the revised "Using a network of temperature lidars to identify temperature biases in the upper stratosphere in ECMWF reanalyses" by Graeme Marlton et al.

General point: please be careful to write in a consistent tense throughout – you mix past and present tense freely in the manuscript. I recommend using the past tense when referring to agreement between datasets at dates in the past.

p.4 l.17. 'The first' needs to be followed by 'the second'

p.6 l.20 Surely the evolution of IFS is due to R+D, not the other way round

p.6 l.28 'significantly **a**ffected **by**'

p.7 l.3 'To ensure that the comparison is accurate, the lidar's geometric height coordinates were first converted to geopotential height'

p.7 l.7 'Vertical resolution **of**'

p.7 l.13 The sentence 'ERA-interim at the points studied here has a cold bias in the region of -3 to -4 K.' is completely inconsistent with fig.2 and should be deleted.

p.7 l.21. 'Both the warm bias near the model top and the cold bias between 1 hPa and 10 hPa are present throughout the year, with the cold bias being strongest at all sites between November and February.'

p.8 l.19 'could explain' would be more accurate. Also would 'difference in the sign of the results' not be better than 'polarity', which is something I associate with magnets?

p.8 l.30 'vertical resolution **of ERA5**'

p.9 l.16 'and 10 hPa**, but not as good as that with the** lidar'

p.9 l.21 'of 3 K, **while** at 1 hPa'

p.9 l.35 original text is better

p.10 l.25 delete 'that the introduction of COSMIC GPSRO showed'

p.10 l.29 'since' rather than 'given'; 'in 1998' rather than '19998'

p.11 l.14 'affected' not 'may have an affect on' ; 'have increased' not 'increase'

p.11 l.23 Replace 'Given …  height' with 'The abrupt and consistent reduction of the warm bias at all sites during 1998 corresponded with the advent of NOAA-15 AMSU-A data, suggesting that assimilation of this data stream caused the reduced bias.'

p.11 l.26 '**was** most dominant'

p.11 l.27 'caused' rather than 'could be the reason for'

p.12 l.3 'was reduced'

p.12 l.6 'appears to have reduced the'

p.12 l.19 'comparisons **between** ERA'

p.12 l.24 '**was** within'

p.12 l.33 what does the word 'saturate' mean here? Please find a more suitable alternative

p.13 l.4 'effect' not 'affect'

p.13 l.14 'reference with which to compare other reanalyses'

p.13 l.23 'have made high vertical resolution measurements for nearly 30 years'

---

## Author Response (AR2)

We thank the Editor for their comments and have made the proposed corrections. Please see the marked up revised manuscript

*Editor's comments on the revised "Using a network of temperature lidars to identify temperature biases in the upper stratosphere in ECMWF reanalyses" by Graeme Marlton et al.*

*General point: please be careful to write in a consistent tense throughout – you mix past and present tense freely in the manuscript. I recommend using the past tense when referring to agreement between datasets at dates in the past.*

The Manuscript has been read and tenses have been made more consistent.

The following corrections have been implemented:

*p.4 l.17. 'The first' needs to be followed by 'the second'*
*p.6 l.20 Surely the evolution of IFS is due to R+D, not the other way round*
*p.6 l.28 'significantly **affected by**'*
*p.7 l.3 'To ensure that the comparison is accurate, the lidar's geometric height coordinates were first converted to geopotential height'*
*p.7 l.7 'Vertical resolution **of**'*
*p.7 l.13 The sentence 'ERA-interim at the points studied here has a cold bias in the region of -3 to -4 K.' is completely inconsistent with fig.2 and should be deleted.*
*p.7 l.21. 'Both the warm bias near the model top and the cold bias between 1 hPa and 10 hPa are present throughout the year, with the cold bias being strongest at all sites between November and February.'*
*p.8 l.19 'could explain' would be more accurate. Also would 'difference in the sign of the results' not be better than 'polarity', which is something I associate with magnets?*
*p.8 l.30 'vertical resolution **of ERA5**'*
*p.9 l.16 'and 10 hPa**, but not as good as that with the** lidar'*
*p.9 l.21 'of 3 K, **while** at 1 hPa'*
*p.9 l.35 original text is better*
*p.10 l.25 delete 'that the introduction of COSMIC GPSRO showed'*
*p.10 l.29 'since' rather than 'given'; 'in 1998' rather than '19998'*
*p.11 l.14 'affected' not 'may have an affect on' ; 'have increased' not 'increase'*
*p.11 l.23 Replace 'Given … height' with 'The abrupt and consistent reduction of the warm bias at all sites during 1998 corresponded with the advent of NOAA-15 AMSU-A data, suggesting that assimilation of this data stream caused the reduced bias.'*
*p.11 l.26 '**was** most dominant'*
*p.11 l.27 'caused' rather than 'could be the reason for'*
*p.12 l.3 'was reduced'*
*p.12 l.6 'appears to have reduced the'*
*p.12 l.19 'comparisons **between** ERA'*

*p.12 l.24 '**was** within'*

*p.12 l.33 what does the word 'saturate' mean here? Please find a more suitable alternative*

*p.13 l.4 'effect' not 'affect'*

*p.13 l.14 'reference with which to compare other reanalyses'*

*p.13 l.23 'have made high vertical resolution measurements for nearly 30 years'*